# An Extensively Hydrolyzed Formula Supplemented with Two Human Milk Oligosaccharides Modifies the Fecal Microbiome and Metabolome in Infants with Cow’s Milk Protein Allergy

**DOI:** 10.3390/ijms241411422

**Published:** 2023-07-13

**Authors:** Claire L. Boulangé, Helle K. Pedersen, Francois-Pierre Martin, Léa Siegwald, Albert Pallejà Caro, Aron C. Eklund, Wei Jia, Huizhen Zhang, Bernard Berger, Norbert Sprenger, Ralf G. Heine

**Affiliations:** 1Nestlé Institute of Health Sciences, Nestlé Research, Société des Produits Nestlé S.A., 1000 Lausanne, Switzerland; francoispierremartin@hotmail.com (F.-P.M.); lea.siegwald@rd.nestle.com (L.S.); norbert.sprenger@rdls.nestle.com (N.S.); 2Clinical Microbiomics, 2100 Copenhagen, Denmark; helle@clinical-microbiomics.com (H.K.P.); apalleja@clinical-microbiomics.com (A.P.C.); eklund@clinical-microbiomics.com (A.C.E.); 3University of Hawaii Cancer Center, Honolulu, HI 96813, USA; weijia1@hkbu.edu.hk (W.J.); hzhang@cc.hawaii.edu (H.Z.); 4School of Chinese Medicine, Hong Kong Baptist University, Kowloon Tong, Hong Kong 999077, China; 5Nestlé Health Science, 1800 Vevey, Switzerland; ralf.heine@au.nestle.com

**Keywords:** human milk oligosaccharides, microbiome, metagenomics, fecal community type, metabolomics, amino acids, bile acids, short-chain fatty acids

## Abstract

Cow’s milk protein allergy (CMPA) is a prevalent food allergy among infants and young children. We conducted a randomized, multicenter intervention study involving 194 non-breastfed infants with CMPA until 12 months of age (clinical trial registration: NCT03085134). One exploratory objective was to assess the effects of a whey-based extensively hydrolyzed formula (EHF) supplemented with 2′-fucosyllactose (2′-FL) and lacto-*N*-neotetraose (LNnT) on the fecal microbiome and metabolome in this population. Thus, fecal samples were collected at baseline, 1 and 3 months from enrollment, as well as at 12 months of age. Human milk oligosaccharides (HMO) supplementation led to the enrichment of bifidobacteria in the gut microbiome and delayed the shift of the microbiome composition toward an adult-like pattern. We identified specific HMO-mediated changes in fecal amino acid degradation and bile acid conjugation, particularly in infants commencing the HMO-supplemented formula before the age of three months. Thus, HMO supplementation partially corrected the dysbiosis commonly observed in infants with CMPA. Further investigation is necessary to determine the clinical significance of these findings in terms of a reduced incidence of respiratory infections and other potential health benefits.

## 1. Introduction

The developing gut microbiome undergoes major changes from birth to early childhood and significantly impacts early immune development [1,2]. Diet is one of the main factors affecting this developmental trajectory. Human milk is a complex biofluid comprising multiple nutritive and non-nutritive components that support early growth, development and immune maturation [3]. Among the non-nutritive components, human milk oligosaccharides (HMO) are a highly abundant and diverse group of complex carbohydrates that make up the third biggest component of human milk solids [4]. HMOs are formed by different combinations of five basic monosaccharides (fucose, N-acetyl-neuraminic acid, N-acetyl-glucosamine, glucose, and galactose) and are considered non-digestible by human digestive enzymes. The vast majority of HMO passes through the gastrointestinal tract (GIT) and drives the establishment and maturation of the early microbiome, largely through providing the specific substrate for an enrichment of infant-type, HMO-utilizing bifidobacteria [5,6]. The saccharolytic fermentation of HMO by bifidobacteria produces metabolites such as acetic acid and aromatic lactic acids which have important immune-modulating functions and enhance the protection against a broad range of infections [7,8,9,10,11]. Due to their similarities with intestinal glycans, HMO can directly interact with the gut epithelial barrier and modify the physical host-microbe interaction, with preventive effects against colonization with enteric pathogens and improved mucosal barrier function [4,12].

A previous clinical trial has shown that non-hydrolyzed cow’s milk-based infant formula supplemented with 2′-fucosyllactose (2′-FL) and lacto-*N*-neotetraose (LNnT) is well tolerated and supports normal growth in healthy term infants [13]. In that study, reduced incidences of lower respiratory tract infections (LRTI) and medication use (antipyretics and antibiotics) were reported in the study group fed with the HMO-supplemented infant formula, compared to the control group receiving no HMO [13]. The reduced antibiotic usage in the HMO-supplemented feeding group was associated with a shift of the gut microbiome composition closer to patterns observed in breastfed infants [14]. Further investigations, using a combination of machine learning and in-vitro experiments, suggested that bifidobacteria (in particular, *Bifidobacterium longum* subsp. *infantis*) were involved in the observed reduction in LRTI [15]. Martin et al. described molecular changes related to HMO feeding and protection from LRTI, including an increase in gamma glutamylation and N-acetylation of amino acids, as well as a decrease in inflammatory lipids. Changes in amino acid and lipid metabolism were linked to *Bifidobacterium* and *Bacteroides* species, respectively [16].

The introduction of a complementary diet from 4 to 6 months of age represents a turning point in the early development of the gut microbiome and immune system. The increased dietary intake of fibers and proteins promotes the establishment of clades that are capable of digesting more diverse glycans and proteins. This is reflected in a progressive shift from a milk-adapted microbiome and metabolism towards a more diverse, adult-like microbial ecosystem rich in firmicutes [17,18]. During weaning, the infant’s immune system is exposed to a wider range of dietary and bacterial antigens, which is thought to stimulate the appropriate maturation of early immune responses and prevent the onset of pro-inflammatory or allergic pathologies [19,20,21].

Early gut microbiome composition and metabolic profiles change significantly after diet diversification commences. Fermentation of dietary fibers and proteins leads to an increase in colonic short-chain fatty acids (SCFA), such as butyric acid and propionic acid, and branched-chain fatty acids (BCFA), respectively [22]. These compounds are plausible mediators of host-microbiome interactions, which modulate gut and immune functions during this period [2,23]. In addition, an inadequate maturation of the microbiome and its associated metabolic profile may affect immune development and functioning later in life [1,2,24,25]. Cow’s milk protein allergy (CMPA) is one of the most common food allergies in infancy and early childhood [26]. The dietary management of young infants with CMPA who are not exclusively breastfed relies on specialized formulas, including extensively hydrolyzed formula (EHF) or amino acid-based formula [27,28]. HMO supplementation of EHF formulae for CMPA may provide beneficial effects on microbiome and immune development.

The primary objectives of our clinical study were to investigate the effects of 2′-FL and LNnT supplementation in infants with CMPA on weight gain, anthropometric measurements, and safety as previously published [29]. Infants were randomized to receive a whey-based, lactose-containing EHF supplemented with 2′-FL and LNnT or an HMO-free control EHF from the time of randomization to 12 months of age. The study showed that infants in both study groups achieved normal growth. Furthermore, the study suggested that the HMO-supplemented formula may reduce the risk of respiratory and gastrointestinal infections in the first year of life [29]. The exploratory objective of this study was to assess the effects of 2′-FL and LNnT on the fecal microbial ecosystem in this population. Thus, stool samples were collected at various time points during this study, and shotgun metagenomics as well as targeted mass spectrometry (MS)-based metabolomic analyses were performed. The results enabled us to characterize HMO-mediated changes in microbial and metabolic profiles over the study period to 12 months of age.

## 2. Results

### 2.1. Cohort Description

The study cohort comprised 194 non-breastfed infants aged between 2 weeks and 6 months (mean age 3.2 months) from 7 countries in Europe and Singapore with symptoms suggestive of CMPA. Participants were randomized to receive either an EHF supplemented with 2′-FL and LNnT (test formula) or the same EHF without HMO (control formula) until 12 months of age. The protein content of the test formula was slightly lower than that of the control formula (2.5 g/100 kcal vs. 2.2 g/100 kcal). The first 4 months of the study were conducted as a growth trial (principal study period). The introduction of a complementary solid diet was allowed from 4 months of age. Infants were followed to 12 months of age (secondary study period). Demographics of the study cohort, clinical details, and study outcomes (growth, tolerability, and safety) were previously published [29].

Of the 194 participants, 151 (79 test, 72 control) completed the 4-month growth trial without major protocol deviations (per protocol cohort). Of these, 142 (71 test, 71 control) remained in the trial until 12 months of age (visit 6). The microbiome and metabolomic analyses were limited to the per-protocol cohort and performed at baseline (visit 0), after 1 month (visit 1), and 3 months (visit 3) of study formula intake, and during follow-up at 12 months of age (visit 6). The microbiome composition and fecal metabolomic signature were strongly influenced by the infant’s age in both feeding groups (Appendix A). When including all subjects in the analysis, we did not observe a consistent HMO-related microbiome and metabolic signature, probably due to the wide age range at randomization (Appendix A). To overcome this, we stratified subjects into an early enrollment (EE) cohort aged ≤90 days at baseline (*n* = 60) and a late enrollment (LE) cohort aged >90 days at baseline (*n* = 72). This stratification allowed reducing the confounding effects of age and complementary diet, as the EE cohort had not started non-formula foods at the time of the baseline visit (V0) and 1-month follow-up (V1). The study design is described in Figure 1.

### 2.2. Human Milk Oligosaccharides Supplementation Shifts Microbiome Development by Favoring a Bifidobacteria-Enriched Microbiome

In both feeding groups, the microbiome development was strongly influenced by age (Appendix A). An increase in phylogenetic diversity was observed over the study period for both treatment groups, which is characteristic of microbiome development during the first year of life (Figure 2A, Appendix A). In the EE cohort, the phylogenetic diversity was significantly lower in the test group than in the control group at 3 months after enrollment (V3, *p* = 0.0011, Cliff’s delta (CD): −0.49, lower confidence interval (LCI): −0.71, upper confidence interval (UCI): −0.20). This trend remained at 12 months of age but was no longer statistically significant (V6, *p* = 0.92). In the LE cohort, a decreased phylogenetic diversity was observed in the test group at 12 months of age (V6, *p* = 0.027, CD: −0.32, LCI: −0.55, UCI: −0.04).

A clustering of microbiome composition using the Dirichlet multinomial mixtures approach was performed to investigate the trajectory of the overall microbiome development. We identified an optimal number of five microbial community types (FCT) at the genus level (Figure 2B). FCT were numbered from FCT1 to FCT5 according to their temporal appearance (Figure 2C). Extensive differences in taxonomic composition and functionality were observed across the FCT (Figure 2B). The youngest infants had the highest proportion of FCT1 (enriched in proteobacteria), while FCT2 (enriched in actinobacteria) was present in higher proportions in infants aged from 2 to 8 months. At 12 months of age, most infants had transitioned to FCT3, FCT4, or FCT5. FCT3 was high in firmicutes and *Lachnoclostridium*. FCT4 was enriched with firmicutes, and FCT5 harbored higher firmicutes and *Faecalibacterium* (Figure 2B,C). The latter also showed a higher capacity to produce butyric acid (Appendix A). The order in which infants transitioned from one cluster to another further characterized the age-related microbiome maturation, with transitions from “early” (FCT1 and FCT2) to “late” (FCT3, FCT4, FCT5) microbial communities (Figure 2C).

When analyzing the FCT composition stratified by visits, feeding groups, and cohorts, significant differences were found between the test and control groups at V3 (Wilcoxon–Mann–Whitney (WMW) *p* = 0.029, CD: −0.32, LCI: −0.56, UCI: −0.022) and V6 (WMW *p* = 0.0098, CD: −0.35, LCI: −0.57, UCI: −0.08) in the EE and LE cohorts, respectively (Figure 2D,E, Appendix A). In both cohorts, the test group had higher proportions of infants with early FCTs. In the EE cohort at V3, FCT3 was less represented in the test group compared to the control group (Fisher’s exact, *p* = 0.025). In the LE cohort, FCT5 was significantly less prevalent in the test group than in the control group (Fisher’s exact, *p* = 0.049) at V6. To further study the effect of HMO supplementation on the development of the infant’s microbiome, we used a Kaplan–Meier approach to compare the duration before transitioning from one FCT to the next between feeding groups (Figure 2F,G). In the EE cohort, the transition to FCT3 (or FCT4 or FCT5) occurred significantly later (log-rank *p* = 0.033, effect size: 0.56, LCI: 0.32, UCI:0.96) in the test group, which indicated that the passage from “early” to “late” FCTs was slowed down by HMO supplementation in this cohort. No significant difference between the feeding groups was found in the LE cohort (Appendix A).

Differences in taxonomic composition between feeding groups at each visit in the EE and LE cohorts were assessed using an enrichment analysis at the metagenomic species (MGS) level (Figure 3, Appendix A). Many *Bifidobacterium* species appeared to be enriched in response to HMO feeding at V1 (Figure 3A) and V3 (Figure 3B) in the EE cohort, although these apparent differences did not reach statistical significance. However, when performing the analysis at the genus level, we observed a significant enrichment of bifidobacteria in the test group, with the largest positive effect size found for an increase in the abundances of *Bifidobacterium* in HMO-treated infants at V1 (WMW, *p* = 0.00047, CD: 0.74) and at V3 (WMW, *p* = 0.0152, CD: 0.51). This enrichment in bifidobacteria was no longer significant at V6 (Appendix A). A similar trend was observed in the LE cohort at V1, although fewer bifidobacterial species were enriched in the test group (Appendix A), while no other taxa at the species or genus level passed the significance threshold. This trend disappeared at V3 and V6.

In summary, these observations show that feeding of the HMO-supplemented EHF promoted an enrichment of bifidobacteria and slowed the microbiome maturation. This effect was stronger in infants who were commenced on the HMO-supplemented EHF before 3 months of age.

### 2.3. Human Milk Oligosaccharides Supplementation Impacts Fecal Metabolites from Colonic Amino Acid and Bile Acid Metabolism in the EE Cohort

A total of 137 metabolites were identified and quantified, including HMO, SCFA, lipids, bile acids, and other host-gut microbial metabolites (Appendix A). Changes in metabolic signatures by feeding group were explored using multivariate and univariate analysis (see Section 4.7). Most metabolic differences between the feeding groups were found at V3 and V6 (Appendix A).

The fecal metabolite profiles combining V1, V3, and V6 samples differed between the EE and LE cohorts. Medium-chain fatty acids commonly found in infant formulae (dodecanoic acid, decanoic acid, and myristic acid) were higher in the EE cohort, whereas lactic acid, acetic acid, and secondary bile acids increased in the LE cohort (Appendix A). These differences are likely explained by age and the introduction of the complementary diet. In the EE cohort, a partial least square-discriminate analysis (PLS-DA) modeled the influence of age and formula type (i.e., control vs. test) along the first and the second component, respectively (Figure 4A). In addition, a Wilcoxon–Mann–Whitney (WMW) test was used to rank metabolites according to their statistical significance between the control and test group at each visit. A total of 14 metabolites differed between the feeding groups in both the multivariate and univariate analyses (Table 1 and Figure 4B). Among them, 2′-FL was significantly higher in the feces of infants in the test group compared to control infants, reflecting HMO supplementation (Figure 4C). Of note, traces of 2′-FL were also found in fecal samples in the control group (average 86 nmol/g at V0, 64 nmol/g at V1, and less than 20 nmol/g at V3 and V6). A decreased fecal content in intermediates of the metabolism of branched-chain amino acids (BCAA), lysine, and aromatic amino acids (AAA) was found in the test group after 3 months of treatment (V3) and maintained to 12 months of age (V6) (Table 1 and Figure 4C). In line with this finding, phenylalanine, a precursor of aromatic compounds, was also reduced at 12 months of age (V6). Together, these results indicate the downregulation of bacterial protein catabolism with HMO supplementation. Furthermore, a decrease in fecal content of a secondary bile acid, dehydrocholic acid, was observed in the test group compared to the control group after 3 months of treatment (V3) but not at 12 months of age (V6). In the LE cohort, we found 10 metabolites discriminating between the control and test groups, with ten free fatty acids (including oleic and palmitoleic acid) being upregulated in the test group at V6 (Appendix A).

### 2.4. Fecal Ratios of Unconjugated/Conjugated Bile Acids and Acetic Acid Levels, Markers of Bifidobacterial Metabolism, Are Modulated by Human Milk Oligosaccharides in the EE Cohort

Bifidobacteria are known to impact the production of SCFA and bile acid profile in the gut microbiome [29,30]. We assessed to what extent HMO supplementation impacted bile acid deconjugation and SCFA production in the study cohort. The unconjugated/conjugated cholic acid (CA/CCA), unconjugated/conjugated chenodeoxycholic acid (CDCA/CCDCA), unconjugated/conjugated lithocholic acid (LCA/CLCA), and total unconjugated/conjugated bile acid ratios (BA/CBA) were calculated. No significant differences in any of the bile acid ratios or SCFA were found between the control and test groups at any visit. However, when looking at the trajectories over time, BA/CBA and CA/CCA significantly decreased in the control group at V1 (WMW, *p* = 0.021 for BA/CBA; WMW, *p* = 0.015 for CA/CCA), V3 (WMW, *p* = 0.0038 for BA/CBA; WMW, *p* = 0.0031 for CA/CCA) and V6 (WMW, *p* = 0.014 for BA/CBA; WMW, *p* = 0.0031 for CA/CCA) compared to the baseline, while these ratios remained unchanged from baseline in the test group (Figure 5A,B). These results suggest that the bifidobacteria-enriched microbiome maintains bile acid deconjugation activity over time in the HMO-supplemented test group. In addition, acetic acid levels in the control group tended to decrease from the baseline to V1 (WMW *p* = 0.09), whereas levels remained stable over this period in the test group (Figure 5C).

### 2.5. Omics Integration Describes Association between Bacterial Function and Fecal Metabolites Involved in Amino Acid and Bile Acids Metabolism

To investigate whether key microbial functions involved in amino acid metabolism and bile acid deconjugation contribute to the changes observed in the fecal metabolome of the EE cohort, the metabolomics data were integrated with the Kyoto Encyclopedia of Genes and Genomes (KEGG) orthologues (KO). The metabolites significantly discriminating between the control and test groups in the EE cohort, as well as the CA/CCA and BA/CBA bile acid ratios, were included in the correlation analysis (*n* = 32 metabolites). Most of these metabolites were fusel acids (isobutyric acid, isovaleric acid, 4-hydroxyphenylacetatic acid, and 2-phenylacetic acid) derived from the fermentation of valine, leucine, tyrosine, and phenylalanine via the Ehrlich or amine pathways (Figure 6A, Appendix A). The KO corresponding to the transaminases, dehydrogenase, and decarboxylases involved in these pathways and the choloylglycine hydrolase involved in bile acid deconjugation were identified using KEGG (*n* = 34 KO) and mapped onto the metagenomic dataset (Figure 6B).

Several statistically significant positive associations were found between a cluster of metabolites, including fusel acids, 4-cresol sulfate, phenylalanine, hydrocinnamic acid and pimelic acid, and KO involved in the Ehrlich pathways (Figure 6B). Furthermore, several of these metabolites and KO were positively associated with later FCT at V3 and were lower in infants fed with HMO-supplemented EHF. On the other hand, the same cluster of metabolites was negatively associated with other KO from the amino acid metabolism, including some KO involved in the amine pathways. These results suggest that the microbiome contributes to the production and excretion of these amino acid metabolites via the Ehrlich pathway rather than the biogenic amine pathway. Modulation of the microbiome composition and metabolism by HMO may lead to a downregulation of the Ehrlich pathway and amino-acid metabolites.

This analysis also revealed that bile acid hydrolase was positively correlated with BA/CBA and negatively correlated with dehydrocholic acid. The bile acid hydrolase activity appeared to be higher in infants from the test group at V6, although the difference was not statistically significant. We also found significant positive associations between BA/CBA, CA/CCA, or bile acid hydrolase and late FCTs at V3.

## 3. Discussion

The pathophysiology of CMPA is complex and multifactorial [30]. Several studies have highlighted a dysbiotic gut microbiome in infants with CMPA characterized by a reduced microbial diversity, a loss of beneficial bacteria such as bifidobacteria, and the presence of opportunistic pathogens [31]. Dysbiosis is thought to play an important role in the disturbance of early immune development and regulation [32]. Compensating for this dysbiosis by HMO intake may have a positive impact on the clinical outcomes of CMPA, including immunological tolerance development to cow’s milk protein (‘outgrowth’). In the present study, we found that HMO supplementation impacts the early fecal microbiome development and modulates bacterial metabolic profiles in EHF-fed infants with CMPA. We demonstrated that HMO supplementation was associated with a significant enrichment in bifidobacteria, slowing the progression to an adult-type microbiome more abundant in firmicutes. The HMO effect on microbiome composition was associated with beneficial effects on several metabolic pathways, including SCFA production (i.e., acetic acid), amino acid degradation, and bile acid conjugation. To our knowledge, this is the first study describing detailed microbiome and metabolome patterns related to EHF feeding with or without HMO in infants with CMPA.

CMPA symptoms may manifest at different ages in infancy; therefore, management with an EHF may commence before or after the introduction of complementary solid foods. In the present analysis of the fecal microbiome, we stratified the cohort into infants enrolled before 3 months of age (EE cohort) and those enrolled after 3 months of age (LE cohort). This stratification aimed to reduce the confounding effects of age and complementary diet. The EE cohort was exclusively formula-fed to about 4 months of age, i.e., from enrollment to the 1-month follow-up visit. We hypothesized that the HMO effects on the gut microbiome and metabolome would be easier to identify before solid foods and dietary fiber were introduced.

Global fecal microbial changes were investigated by computing phylogenetic alpha diversity. Using a classifying-based analytical approach, we defined five clusters of samples, i.e., fecal community types (FCT), characterized by similar taxonomic groups at the genus level. Overall, we observed a more diverse and complex gut ecosystem as a function of age which is in accordance with the gut microbiome maturation trajectory described by others [1,2]. The prevalence of early FCTs depicting an ecosystem adapted to a milk-based diet (rich in *Escherichia*, *Klebsiella*, *Veillonella,* and *Bifidobacterium*) is typical of early infancy up to about 4 months. With the introduction of complementary feeding, FCTs evolve toward a more diverse and adult-like ecosystem rich in *Lachnoclostridium*, *Bacteroides,* and *Ruminococcus*.

Our FCT model showed that the intake of HMO slowed the developmental progression toward a mature, adult-type microbiome composition, illustrated by a reduced microbial diversity and an enrichment of early-stage FCTs in the HMO group compared to the control group.

In the EE cohort, these ecological differences were particularly visible after 3 months of HMO intake (V3). A survival analysis showed that, at the same age, fewer infants in the HMO group transitioned to a later FCT stage compared to control infants. The HMO group exhibited enrichment in bifidobacteria and depletion in proteobacteria (e.g., *Escherichia coli*) after 1 and 3 months of HMO formula feeding, thus partially reversing the gut microbial dysbiosis. Within the genus *Bifidobacterium*, *B. longum* subsp. *Infantis, B. breve*, *B. bifidum*, and *B. longum* subsp. *longum*, known to be the main HMO-utilizing taxa associated with breastfeeding, were among the most enriched species after 1 month of HMO EHF feeding compared to controls [10].

In the LE cohort, in which infants were already exposed to solid food before enrolment, HMO-mediated changes in the gut microbiome were less pronounced. Significant differences in alpha diversity and FCT distribution between feeding groups could only be demonstrated at 12 months of age, suggesting that a longer intervention may be required to induce an effect in this population. In the LE cohort, bifidobacteria abundance tended to increase in the HMO group already after 1 month, although this trend did not pass the significance threshold at the species or genus level and disappeared at later visits. Several aspects of dietary intake may explain these observations. The frequency of formula feeding is generally reduced with age; therefore, the overall formula intake and HMO exposure are lower in infants in the LE cohort. Intake of dietary fibers from the complementary diet is likely to confound the HMO effect due to an increased capacity of the microbiome to ferment fiber. As a result, the effects of HMO may be less clear in the LE cohort compared to infants who received the HMO-supplemented EHF before 3 months of age as part of exclusive formula feeding.

Exclusively breastfeeding for 4 to 6 months is known to decrease fecal microbial diversity, delay the maturation of the gut microbiome, and promote the enrichment of HMO-utilizing-bifidobacteria [5]. The latter mediates beneficial health outcomes, including the support of early immune maturation, protection against infection, and potentially a lower incidence of allergic manifestations [13,33,34]. In recent systematic reviews, a low abundance of *Bifidobacterium* during the first months of life was described as a consistent feature related to the risk of developing allergies [35,36]. In addition, bifidobacteria act as a symbiotic contributor to microbial colonization through the production of metabolites, such as acetic acid, which cross-feed other beneficial microbes, including butyrate producers [34,37]. In our study, there were no significant differences in acetic acid concentrations between feeding groups, but HMO feeding for 1 month maintained higher fecal acetic acid levels compared to baseline levels, while it tended to decrease in the control group.

In our cohort of infants with CMPA, HMO feeding was able to reconstitute microbiome patterns closer to those seen in breastfed infants, especially when HMOs were introduced during exclusive feeding with the HMO-supplemented formula. These findings are consistent with other studies in healthy-term infants. Berger et al. reported that formula-fed infants receiving 2′-FL and LNnT exhibited higher levels of *Bifidobacterium* and lower levels of *Escherichia* and unclassified *Peptostreptococaceae* than control formula-fed infants at 3 months of age, with the overall microbial composition being closer to breastfed infants [14]. Dogra et al. identified that enrichment of *B. longum* subsp. *infantis* and increased levels of bifidobacteria-derived acetic acid were factors contributing to the protective effect of 2′-FL and LNnT against LRTI in infancy [15]. Additional metabolic pathways have been identified, including gamma-glutamylation and acetylation of amino acids, altered by the HMO formula feeding [16]. In our study, the microbial ecosystem of the EHF-fed infants with CMPA likely differed from that of healthy infants. In addition, since different protein content can affect the gut microbial ecosystem [38], the microbiome-mediated effect of HMO on infant health may be different in the context of EHF and reference formulae. Nevertheless, we showed that HMO supplementation had a positive impact on the gut microbiome and the gut microbial metabolism of EHF-fed infants with CMPA. These findings suggest that HMOs are critical for the developmental maturation of the gut microbiome in healthy infants, as well as a restoration of a *Bifidobacterium*-depleted, dysbiotic gut microbiome in infants with CMPA.

In the EE cohort, HMO intake was negatively associated with fecal metabolites derived from the bacterial oxidative catabolism (Ehrlich pathway) of branched-chain amino acids (BCAAs) and aromatic amino acids (i.e., isobutyric acid, isovaleric acid, phenylacetic acid, 3,4-hydroxyphenylacetic acid, and 4-cresol sulfate). These reactions are characterized by the oxidation of the amino acid as an electron donor into a volatile carboxylic acid. Oxidative metabolisms provide energy in the form of adenosine triphosphate (ATP) through substrate-level phosphorylation and are often coupled with reductive metabolic pathways to maintain the redox balance [37,39,40]. As our targeted analytic panel did not include the reductive amino acid metabolites, we were unable to assess if the reductive amino acid metabolism was equally affected by HMO-supplemented EHF feeding. Nevertheless, these results indicate the downregulation of energy-forming amino acid catabolism. There is compelling evidence that the presence of fermentable carbohydrates, including HMO, in the proximal colon reduces bacterial amino acid catabolism and instead promotes biosynthetic pathways to meet bacterial requirements for organic N-containing compounds [41,42]. It is also important to note that the dietary protein content in the test group was slightly lower than in the control group (2.5 g/100 kcal vs. 2.2 g/100 kcal), which may have contributed to fewer amino acids reaching the lower gastrointestinal tract and being metabolized by the microbiome.

Feeding an HMO-supplemented EHF reduced the abundance of several amino acid fermentation catabolites, particularly in the EE cohort. Importantly, some metabolites, such as phenols, ammonia, and hydrogen sulfide, can be toxic to colonocytes and may contribute to mucosal inflammation [43,44]. For example, the uremic toxins 4-cresol and 2- or 3-hydroxyphenylacetic acid formed via bacterial tyrosine and phenylalanine fermentation can have negative mucosal and systemic effects [45,46]. A reduction in fecal toxic metabolites may therefore contribute to maintaining mucosal and systemic immune balance.

We found evidence of HMO-mediated metabolic remodeling of bile acid profiles, which was more marked in the EE cohort. Bile acids play a central role in nutrient digestion, fat absorption, and cholesterol metabolism. Primary bile acids are synthesized from cholesterol, conjugated with glycine or taurine in the liver, and secreted into the bile before entering the enterohepatic circulation. A large proportion (80–90%) of the intestinal bile acid pool is reabsorbed in the ileum [47]. A small amount of unabsorbed conjugated primary bile acids enters the large intestine and is further transformed by bacteria before being reabsorbed in the colon. Bacterial bile salt hydrolase (BSH) activity removes the glycine and taurine conjugate and produces unconjugated bile acids. This allows other bacteria to convert these primary bile acids into secondary bile acids by 7α-dehydroxylation [48]. The microbiome is still immature during early infancy, and primary bile acids are maintained at high levels. An increase in microbial diversity and richness which is commonly observed from 6 months of age, is associated with an increase in secondary bile acids [47,49]. In our study, bacterial bile acid deconjugation was maintained at a stable level over time in the HMO group, while it decreased in the control group, suggesting an HMO-mediated upregulation of BSH activity. BSH expression is found across all major phyla, nevertheless, a greater abundance of bifidobacteria has been linked with an upregulation of BSH activity and increased levels of deconjugated bile acids [50,51]. The increased expression of bacterial BSH activity may be associated with beneficial effects on inflammation, hypercholesterolemia, and digestive function, as well as a reduced incidence of *Clostridiodes difficile* infection and symptomatic atopic dermatitis [52,53,54,55].

Not all prebiotics and types of fiber have the same effect on bile acid conjugation and related outcomes. In contrast to HMO, feeding of inulin has recently been shown to increase microbial BSH-derived unconjugated bile acids and promote type 2 inflammation [56], while specific secondary bile acids, such as 3β-hydroxydeoxycholic acid, increased regulatory T-cell populations [57]. Further studies are required to delineate the effect of BSH and specific secondary bile acids on mucosal immunity. In our study, we found a lower fecal concentration of the secondary bile acid dehydrocholic acid after 3 months of HMO-supplemented EHF feeding. Dehydrocholic is produced by oxidation of the hydroxyl group of cholic acid and produces various secondary bile acids [58]. It is unknown whether lower concentrations of secondary bile acids in stool reflect a higher re-uptake or lower production. Primary and secondary bile acids are signaling molecules activating two classes of receptors, the nuclear receptors and the G protein-coupled receptors. The role of bile acids in maintaining energy homeostasis in the liver and regulating hepatic diseases, cardiovascular diseases, or inflammatory bowel disease has been reported [48]. Growing evidence also shows that bile acids may serve as a mediator of inflammation and allergic diseases. Infants with food allergies exhibited an altered profile of secondary bile acids in comparison to infants suffering from asthma [20,59]. Secondary bile acids and derivatives were found to regulate multiple T cell responses by activating, for instance, the retinoic acid signaling in mucosal dendritic cells [20]. Changes in the processing of the bile acids by the microbiota may be an avenue to manage CMPA symptoms. However, the exact mechanisms of actions linking bile acids and allergic manifestations remain to be elucidated. Plasma bile acid profiling is suggested for future studies investigating the role of specific secondary bile acids in infancy. Of note, an age or developmental maturity adapted microbial-infant metabolic crosstalk seems critical for an effective immune competence later in life infant [60].

The present analysis of the microbiome and metabolome has several limitations. Due to a relatively small number of stool samples from study subjects, the statistical power of the analysis was limited. As subjects were enrolled at different ages, the high variability in age at each time point limited the ability to identify HMO effects due to confounding by age and complementary diet. This issue was partially mitigated by stratifying the population in EE and LE cohorts. However, this stratification further limited the statistical power. Nevertheless, we identified a robust HMO-mediated microbial signature that was consistent with existing literature and in accordance with the changes in the fecal metabolic profile, particularly in the EE cohort. The metabolomic analysis was carried out using a targeted approach that allowed us to accurately quantify metabolites from a variety of pathways. However, the assessment of amino acid metabolism was limited as some metabolites were not included in the initial selection of target metabolites.

In conclusion, the microbiome analysis in the present study of infants with CMPA demonstrated that the supplementation of a whey-based EHF with 2′-FL and LNnT enriched the microbiome with HMO-utilizing bifidobacteria and slowed the progression of the microbiome composition towards an adult-type pattern. HMO supplementation partially reversed the dysbiosis common to infants with CMPA and shifted the microbiome composition closer to a pattern typical of breastfed infants. Specific HMO-mediated changes in fecal amino acid degradation and bile acid conjugation were identified in this cohort, with the greatest effect seen in infants who commenced the HMO-supplemented EHF before 3 months of age. The clinical significance of these observations in the context of a reduced incidence of infections and potentially other health benefits requires further investigation.

## 4. Materials and Methods

### 4.1. Study Design and Participants

This controlled, double-blind, randomized, multicenter, interventional clinical trial of two parallel formula-fed groups (Registration: NCT03085134) was conducted to assess the noninferiority in weight gain per day in infants with CMPA being fed an extensively hydrolyzed formula (EHF) supplemented with two human milk oligosaccharides (HMO), compared to that of a control EHF formula. Full-term, non-breastfed infants aged 0–6 months with physician-diagnosed CMPA were enrolled. Further details of the study design and population were published by Vandenplas et al. [29].

### 4.2. Interventions

Infants were randomized to either receive a commercially available 100% whey-based EHF with a protein content of 2.47 g/100 kcal and without HMO (Althéra^®^, Nestlé Health Science, Vevey, Switzerland) or a similar test formula with a reduced protein content of 2.20 g/100 kcal and supplemented with 2′-FL (target concentration: 1.0 g/L) and LNnT (target concentration: 0.5 g/L). As described by Vandenplas et al., the HMO-containing infant formulae had a slightly lower protein content in line with the recent developments in infant formula design which aim to reduce the risk of excess weight gain [29]. Both formulas contained lactose at 3.8 g/100 mL reconstituted formula (about 52% of total carbohydrates) and had a similar micronutrient composition. Treatment allocation was blinded, and both formulas were indistinguishable by taste and appearance.

### 4.3. Stool Collection

Stool samples were collected from 132 infants at baseline (V0), 1 month (V1, *n* = 122), and 3 months (V3, *n* = 120) from enrollment, as well as at 12 months of age (V6, *n* = 116). Varying numbers of samples per timepoint were analyzed due to early withdrawal from the trial or missing samples. Due to the large age heterogeneity at baseline, infants were stratified into early enrollment (EE; aged ≤90 days at baseline; *n* = 60) or late enrollment (LE; aged >90 days at baseline; *n* = 72). Sample sizes of the EE and LE cohorts and time points are summarized in Appendix A.

### 4.4. Fecal DNA Extraction and Ecological Measures

Microbial DNA was extracted from frozen feces, purified, and shotgun sequenced with 2 × 150 bp sequencing, as described previously [61,62,63] (median reads/sample: 21 × 10^6^, range: 10.8 × 10^6^–38.1 × 10^6^). Taxonomic relative abundances were calculated using the metagenomic species (MGS) approach, which enables quantification of both known characterized and uncharacterized microbial species [64]. Full details are outlined in the Appendix A [65,66]. Beta diversity as Bray–Curtis dissimilarity was calculated using the vegan R package. A phylogenetic tree connecting the MGSs was generated using previously identified conserved genes (Appendix A) [67,68]. Alpha diversity as Faith’s phylogenetic diversity (PD) [69,70] was calculated using this tree with the picante R package. All distances and alpha diversity measures were calculated using rarefied abundances (see details in Appendix A) [71,72,73,74,75,76].

### 4.5. Fecal Community Type Clustering and Visualization

Fecal community types (FCT) clustering of all 481 samples were generated by Dirichlet multinomial mixture (DMM) modeling [77], using non-rarefied genus-level counts with the Dirichlet multinomial R package, which assigns each sample to exactly one FCT. DMM models were fitted using two to ten components, and the analysis was repeated ten times. The optimal clustering was reached by a model using 5 components, selected based on the minimal Laplace approximation to the model evidence. The resulting 5 clusters at the genus level (genus cluster, FCT) were named to imply their order or progression (FCT1 > FCT2 > FCT3 > FCT4 > FCT5).

The temporal development of the infant’s microbiome was tracked with a transition model of the FCT clusters by an approach adapted from Stewart et al. [2], with modifications to accommodate the different sampling schedules of this trial. Each sample was assigned to an age group (rounded to the nearest month). In contrast to Stewart et al., we included all data points, even if two consecutive samples were in the same age group or skipped over an age group. Where only transitions from one age group to the immediate next age group were included (i.e., from t = i to t = i + 1), transitions from one age group to any of the following age groups were also included (i.e., from t = i to t > i + 1; e.g., from t = 2 months to t = 4 months). Furthermore, infants were allowed to be considered twice for a given age group to accommodate cases where the difference between the V0 and V1 samples was <30 days (resulting in self-loops).

### 4.6. Statistical Analysis of Microbiome Data

All statistical tests were run using R software (v. 4.0.3, R Core Team (2022)). Charts were generated using the ggplot2, ggrepel, pheatmap, ComplexHeatmap, survminer, and igraph R packages.

Alpha diversity indexes and taxonomical relative abundances were compared between treatment groups using two-sided Wilcoxon–Mann–Whitney tests and Cliff’s delta as effect size, using the wilcox.exact function from the exactRankTests R package and the cliff.delta function from the effsize R package, respectively.

To investigate the time to reach a given or later evolutionary microbiome state, the time of the event was defined as the age at the earliest visit where the infant was observed to have the FCT in question or a later FCT. As many infants were not observed in all clusters (due to in-frequent sampling or biology), the time of the event was defined as the age at the earliest visit where the infant was observed to have the FCT3 or later FCT, e.g., for ‘Proportion in FCT2 or lower’, time of event would be the earliest visit where the infant had an FCT3, FCT4, or FCT5 community type. Differences in FCT transition probabilities between treatment groups were tested for FCT2, FCT3, FCT4, and FCT5 using Cox proportional hazards regression and global statistical significance reported by the score log-rank test using the survival R package. Data were right- and interval censored, where the intervals between visits were long and of variable length (especially towards the end), resulting in uncertain estimates of the actual time of progression from one cluster to the next. The seven infants with only a V6 sample were excluded from this analysis.

Finally, a taxon set enrichment analysis (TSEA) was performed at the family and genus level (full details are outlined in the Appendix A).

### 4.7. Metabolomics of Fecal Samples and Statistical Analysis of Metabolomics Data

Targeted metabolomics analysis was performed on stool samples of 84 infants (control mean age: 102 days, test mean age: 104 days at baseline) collected at V0, V1, V3, and V6 to measure bile acids, other host-gut microbial metabolites, lipids and HMO. The samples were extracted and prepared according to previously published methods [78,79,80]. Information on chemicals and reagents is detailed in Appendix A. For the bile acid analysis, the calibration curve samples were prepared in the blank matrix and processed in the same way as real biological samples. Ultra-performance liquid chromatography coupled to tandem mass spectrometry (UPLC-MS/MS) system (ACQUITY UPLC-Xevo TQ-S, Waters Corp., Milford, MA, USA) was used to quantitate 43 bile acids based on previously published protocols [78,81]. Data acquisition was performed using MassLynx version 4.1, and bile acid quantification was performed using the TargetLynx applications manager version 4.1 (Waters, Milford, MA, USA). For other host-gut microbial metabolites and lipid analysis, samples were analyzed using a previously published targeted metabolite assay method using UPLC-MS/MS [62]. This second method enabled the quantification of a total of 91 metabolites, including amino acids and derivatives, carboxylic acids, fatty acids, hydroxy acids, keto acids, aromatics, and lipids. For the quantitation of HMO, including 2′-fucosyllactose, lacto-*N*-neotetraose, free lactose, and free fucose, samples were analyzed according to a previous method with minor modification using UPLC-MS/MS [80].

Metabolomic data for the EE and LE cohorts were analyzed separately using a multivariate approach in SIMCA (SIMCA-P16, SARTORIUS STEDIM, Umea, Sweden). Partial least squares regression (PLS) and discriminant analysis (PLS-DA) were employed to model either age, treatment effect, or both at each time timepoint (V1, V3, or V6) or at combined time points (V1,V3,V6 or V3,V6). The model robustness was evaluated based on the value of the cross-validation parameter (Q2Y) and permutation tests. The model predicting age and treatment and combining V3 and V6 time points was the most robust and allowed us to maximize the metabolic differences between the control and test groups. The variable importance in projection (VIP) score was calculated for each feature. The VIP sore corresponds to the weighted sum of squares of the PLS weights, considering the amount of explained Y-variance in each component. The variable with a VIP > 1 is considered significant as its influence on the explanation of the outcomes is above average.

Univariate Wilcoxon–Mann–Whitney (WMW) tests on metabolites were also performed at each time point. Due to the low sample size and the exploratory nature of this study, no adjustment for the false discovery rate was applied. The metabolites were considered the most important in the treatment group discrimination if they had VIP score > 1, |PLS coefficient (*p*(corr))| > 0.2, and WMW *p*-value < 0.05.

To test the hypothesis that HMO supplementation exerts an effect on fecal bile acids and SCFA content over time, the differences in the abundance of SCFA and unconjugated/conjugated bile acid ratio between a given time point (V1, V3, or V6) and baseline (V0) were evaluated using WMW tests in the control and the test group individually.

### 4.8. Metagenomics—Metabolomics Data Integration

Metabolites are significantly different between control and HMO at V3 or V6, and bile acid ratios affected by HMO supplementation over time were selected (*n* = 16). KOs corresponding to the key enzymes involved in the same metabolic pathways (i.e., BCAA, aromatic amino acid, and bile acids metabolism) were identified using the KEGG database (version 102.0) and mapped in our dataset.

Interdomain correlation between KO and metabolite abundances: The analysis was based on all V3 and V6 samples from EE infants with available metagenomic and metabolomic information (*n* = 70). KOs and metabolites that were detected in <3 of the included 70 samples were filtered out. Pairwise interdomain correlations between the normalized abundances of the remaining 17 KOs and 16 metabolites were evaluated by Kendall correlation (two-sided) using the Kendall R package. The Benjamini–Hochberg procedure was used to control the false discovery rate at 10% (concurrently on the entire matrix of all 272 pairwise combinations of metabolites and KOs). MGSs and metabolites with at least one significant correlation (FDR < 10%) were displayed in a heatmap (Figure 5). The rows and columns of the heatmap were ordered based on hierarchical clustering of the Kendall correlation coefficient (Tau) values using Euclidian distance as a dissimilarity measure and Ward clustering.

Association between metabolites/KOs and HMO treatment: For the V3 and V6 visits separately, each of the 16 metabolites and 17 KOs was compared between HMO-treatment groups (control vs. test) with two-sided Mann–Whitney U tests and Cliff’s delta as effect size. Each comparison was based on all available samples from EE infants for the given visit (i.e., KO comparisons included samples without matching metabolomic information). The Benjamini–Hochberg procedure was used to control the false discovery rate at 10% for each visit individually.

Association between metabolites/KOs and FCT clusters: For the V3 and V6 visits separately, each of the 16 metabolites and 17 KOs was tested for association with progressive FCT cluster distribution with Kendall correlation, where the FCT clusters were encoded as ordered factors, with FCT1 = 1, FCT2 = 2, and so on. Each comparison was based on all available samples from EE infants for the given visit (i.e., KO comparisons included samples without matching metabolomic information). The Benjamini–Hochberg procedure was used to control the false discovery rate at 10% for each visit individually.

## Figures and Tables

**Figure 1 ijms-24-11422-f001:**
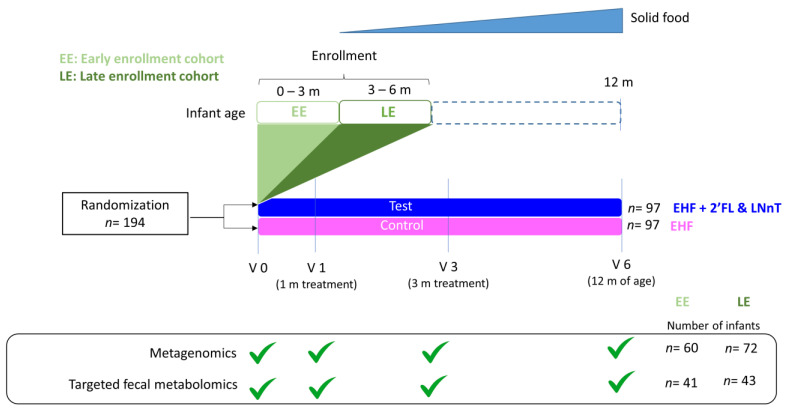
Description of study design: One hundred ninety-four infants were randomized to a test group fed EHF supplemented with 2′-FL and LnNT and a control group fed the same EHF without HMO. Due to the large age heterogeneity at enrollment (see Appendix A), the population was divided into an early enrollment (EE: aged 0 to 3 months) and a late enrollment cohort (LE: aged 3 to 6 months). Fecal samples were collected at 1 month (V1) and 3 months from the start of the study formulas (V3), as well as at 12 months of age (V6). If sufficient sample volume was available, metagenomics and targeted metabolomics analysis were performed. In total, samples from 132 infants (EE: *n* = 60; LE: *n* = 72) were available for metagenomic analysis, and 84 samples for fecal metabolomics (EE: *n* = 41; LE: *n* = 43), respectively. More details about sample numbers in the control and test groups at baseline (V0), visit 1 (V1), visit 3 (V3), and visit 6 (V6) are available in Appendix A.

**Figure 2 ijms-24-11422-f002:**
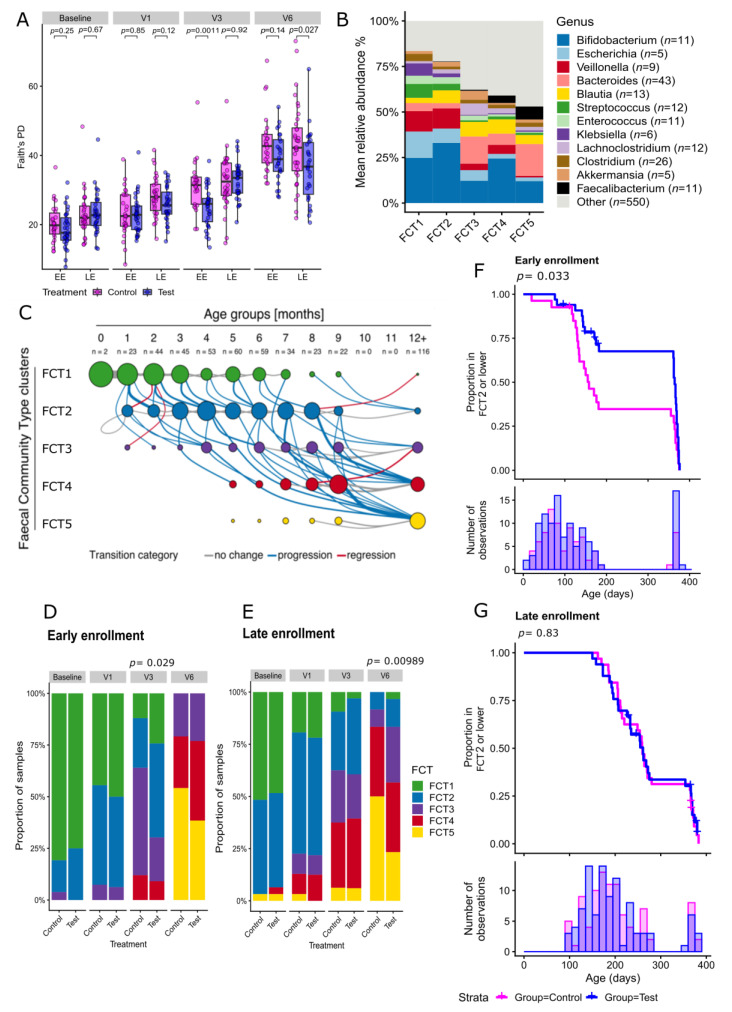
Comparison of microbiota compositions and trajectories between feeding groups (test vs. control), stratified by early enrollment (0–3 months of age; EE) and late enrollment (3–6 months of age; LE). (**A**) Alpha diversity (Faith’s phylogenetic diversity [PD]) of the gut microbiomes of the infants in the two feeding groups at each timepoint (V0, V1, V3, V6) stratified for the EE and LE cohorts. Box plots show the median, 25th and 75th percentiles with Tukey whiskers. At each time point, the two treatment groups were compared, stratified by EE and LE, using a two-sided Mann–Whitney U test. Nominal *p* values are indicated above the groups. (**B**) Taxonomical overview of the 5 fecal community types (FCT) at the genus level. Bar plots display the mean abundance within each FCT of the 10 most abundant taxa; the remainder are grouped in “Other.” (**C**) The transition model analysis depicts the temporal progression from early to late FCT clusters based on all 481 samples. Node sizes represent the fraction of infants in a given cluster per age group (column). Line widths represent the fraction of transitions per age group (column). The line color indicates the transition category (grey: no change, blue: progression, red: regression). In analogy to the methodology used by Stewart et al., only transitions with frequencies >4% are shown [2]. (**D**,**E**) Distribution of FCTs per feeding group in the EE (**D**) and LE cohort (**E**). For each time point, the proportion of samples assigned to each of the five FCTs is shown, stratified by treatment group and enrollment cohort. (**F**,**G**) Kaplan–Meier plot illustrating the transitioning to FCT3 (or later FCT, i.e., FCT4 or FCT5) as a function of age for the EE cohort (**F**) and LE cohort (**G**). Time of event is defined as the age at the earliest visit where the infant is observed to have the FCT3 or later FCTs. The two treatment groups were compared using the log-rank test, and nominal *p* values are indicated above each plot. Vertical lines indicate censored data. The seven infants with only a V6 sample are excluded. The bottom panel shows the number of samples for a given age (bin width = 15 days) and illustrates the uneven age distribution. NB: The LE cohort has no observations during the first 90 days. A cross on the survival line is marked when data are no longer available for a given infant beyond that time point.

**Figure 3 ijms-24-11422-f003:**
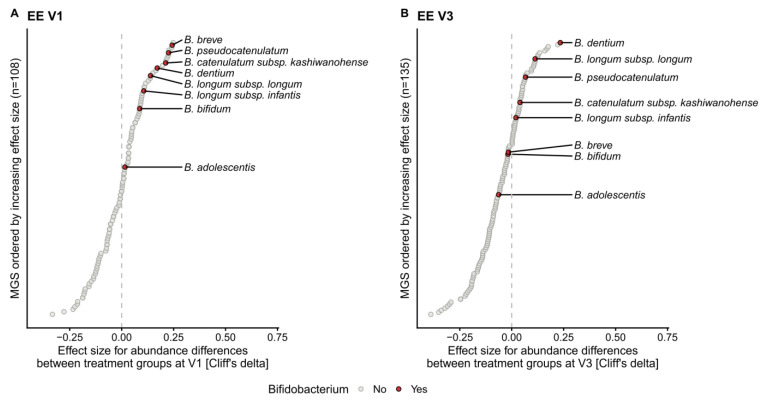
*Bifidobacterium* species enrichment in the test group at V1 and V3 in the EE cohort. Each circle indicates the effect size (Cliff’s delta) of a single metagenomic species (MGS) in comparison to the relative abundance between the control and HMO-treated infants at (**A**) V1 and (**B**) V3. The included MGS (y-axis) are sorted by their corresponding Cliff’s delta (x-axis). The following numbers of samples were included at V1: EE control, *n* = 27; EE test, *n* = 32; and at V3: EE control, *n* = 25; EE test, *n* = 33.

**Figure 4 ijms-24-11422-f004:**
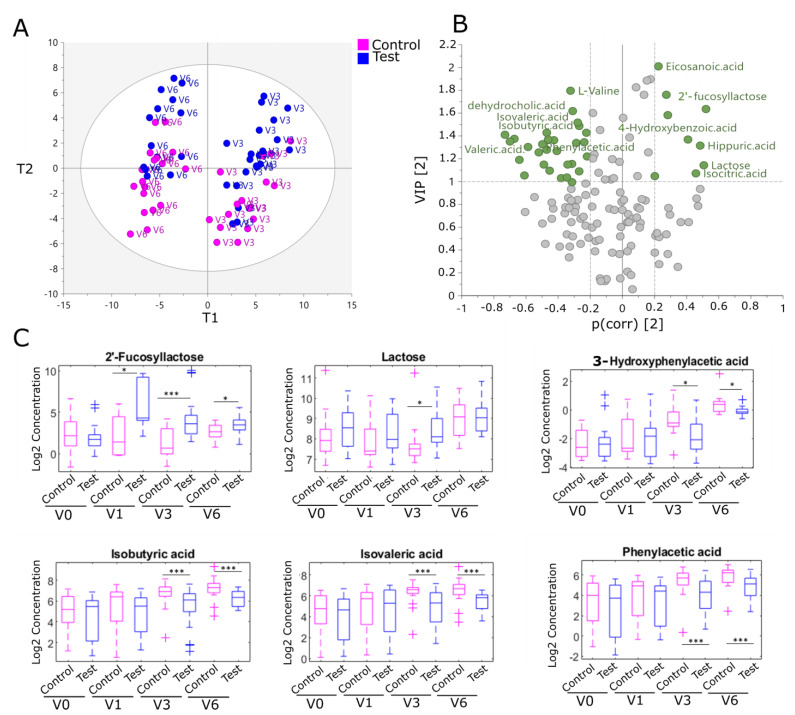
HMO supplementation impacts the fecal metabolic profile in the EE cohort. (**A**) PLS regression score plot modeling age on the first component and feeding groups on the second component at V3 and V6. (**B**) Volcano plot showing cumulative VIP score and correlation coefficient to treatment (PC2). Each dot corresponds to a metabolite. Most discriminating metabolites (VIP > 1 & |PLS coefficient (*p*(corr))| > 0.2) to feeding groups are highlighted in green. (**C**) log_2_ scaled concentration (nmol/g) of 2′-FL, lactose, 3-hydroxyphenylacetic acid, phenylacetic acid, isobutyric acid, and isovaleric acid in the control (highlighted by pink box plots) and test group (highlighted by blue box plots) at each visit. The cross symbols show the outliers that are 1.5 times the interquartile range away from the bottom or top of the box. Significant differences between treatment groups were calculated using the Wilcoxon–Mann–Whitney test (*p*-value * <0.05, *** <0.001).

**Figure 5 ijms-24-11422-f005:**
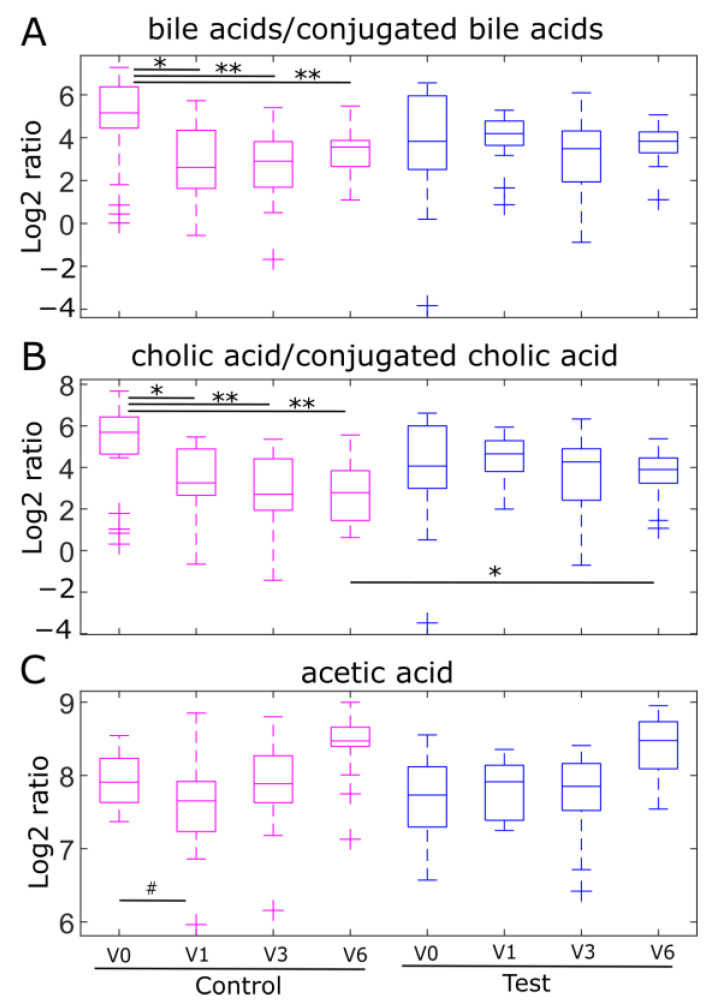
HMO supplementation maintains fecal unconjugated/conjugated bile acid ratios and fecal acetic acid levels at each study visit. (**A**) The trajectory of the total unconjugated/conjugated bile acid ratio (total unconjugated bile acids is the sum of cholic acid, chenodeoxycholic acid, lithocholic acid, and deoxycholic acid). The total conjugated bile acids are the sum of taurocholic acid, taurochenodeoxycholic acid, taurodeoxycholic acid, glycocholic acid, taurolithocholic acid, glycochenodeoxycholic acid, and glycodeoxycholic acid). (**B**) Unconjugated/conjugated cholic acid ratio (**C**) Acetic acid in the control group and test group at each visit. Control group at each timepoint is presented in pink and test group is presented in blue. Significant differences between visits or between treatment groups were calculated using the Wilcoxon–Mann–Whitney test (*p*-value # <0.1 * <0.05, ** <0.01).

**Figure 6 ijms-24-11422-f006:**
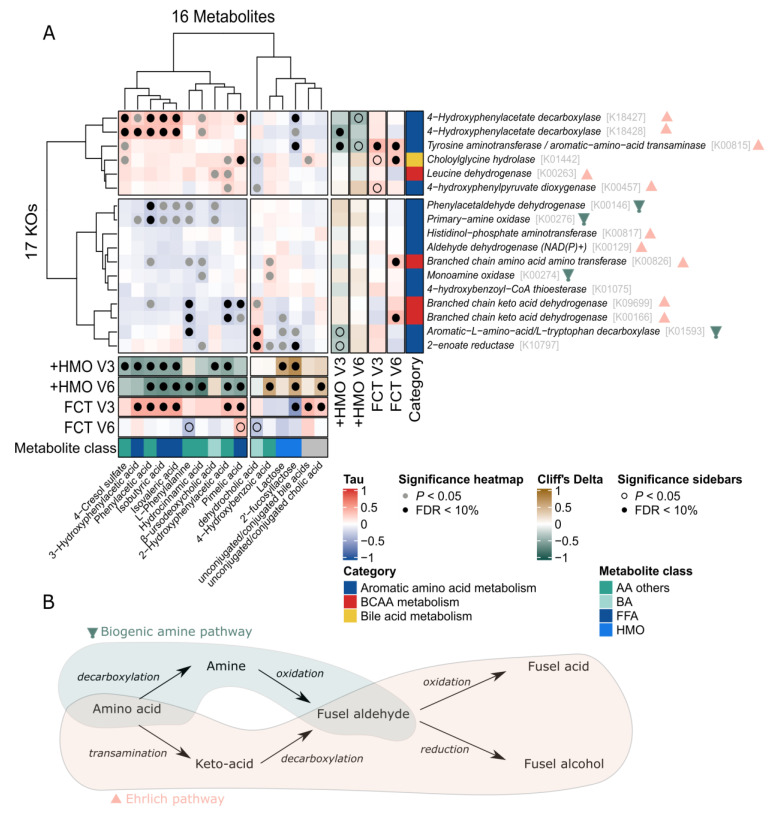
The fecal metabolome reflects HMO-related changes in the bacterial proteolytic activity (**A**) Heatmap illustrating the correlation between KEGG Orthologs (KO) and stool metabolites. Association map of the analyses integrating FCT distribution, test formula effect, the gut microbiome, and the stool metabolome in the early enrollment (EE) cohort at V3 and V6. The main ‘heatmap’ panel shows Kendall’s correlation between a priori selected bacterial enzyme (KOs) and stool metabolites using all V3 and V6 samples with matched metagenomic and metabolomic information (samples, *n* = 70). The colors indicate the direction and magnitude of the correlation (Kendall’s Tau correlation coefficient), where red means a positive correlation between the KO and the stool metabolite, and blue means a negative correlation. Statistically significant correlations are indicated with filled grey (Kendall correlation *p* < 0.05) or black (Kendall correlation FDR < 10%) circles. The right and bottom ‘sidebar’ panels show associations between the same KOs and metabolites, respectively, and HMO-treatment groups (+HMO) or correlation with FCT distribution (FCT) at V3 and V6. For +HMO, the colors indicate the direction and magnitude of the association (Cliff’s delta). Brown means that the KO or metabolite is more abundant in HMO-treated infants, and green means that the KO or metabolite is more abundant in control infants. For FCT, the colors indicate the correlation (Kendall’s Tau correlation coefficient), where red means that the KO or metabolite is enriched in late FCT, and blue means it is enriched in early FCT. Statistically significant associations/correlations are indicated with open (MWU/Kendall correlation *p* < 0.05) or filled (MWU/Kendall correlation FDR < 10%) circles. For sidebars, the following number of samples were included: At V3/V6, associations with KOs, control, *n* = 25/24; test, *n* = 33/25; associations with metabolites, control, *n* = 17/18; test, *n* = 21/14 (detailed statistics in Supplementary Appendix A). (**B**) Production of fusel acids and fusel alcohols from amino acids via the Ehrlich or the biogenic amine pathway.

**Table 1 ijms-24-11422-t001:** Summary of the main metabolites discriminating between the control and test groups in the EE cohort at V3 and V6.

	PLS Model at V3 and V6		WMW Test V3		WMW Test V6	
Metabolites	Correlation with Treatment	VIP Score	FC Test/Control	*p* Value	FC Test/Control	*p* Value
2′-Fucosyllactose	0.52	1.63	19.07	0.0001	2.44	0.0200
Lactose	0.50	1.14	1.80	0.0033	0.98	0.5600
Hydroxybenzoic acid	0.41	1.37	1.01	0.8100	2.34	0.0140
2-Hydroxyphenylacetic acid	−0.58	1.30	0.31	0.0100	0.60	0.054
3-Hydroxyphenylacetic acid	−0.28	1.52	0.24	0.0040	0.77	0.011
Phenylacetic acid	−0.70	1.35	0.24	0.0089	0.34	0.0065
Hydrocinnamic acid	−0.46	1.37	0.21	0.1400	0.25	0.0007
4-Cresol sulfate	−0.31	1.00	0.28	0.0320	0.40	0.14
L-Phenylalanine	−0.34	1.14	1.31	0.3700	0.41	0.0083
Isobutyric acid	−0.67	1.37	0.44	0.0098	0.36	0.0035
Isovaleric acid	−0.73	1.41	0.28	0.0080	0.44	0.00730
Pimelic acid	−0.22	1.43	0.58	0.1230	0.57	0.0210
Dehydrocholic acid	−0.30	1.62	0.51	0.0490	0.69	0.1800

PLS: Partial Least Squares regression VIP:Variable influence on projection FC: Fold change WMW: Wilcoxon Mann Whitney.

## Data Availability

Data are available on request from the Chief Science & Medical Officer, Nestlé Health Science, 1800 Vevey, Switzerland.

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
