# Peer review of "An Extensively Hydrolyzed Formula Supplemented with Two Human Milk Oligosaccharides Modifies the Fecal Microbiome and Metabolome in Infants with Cow’s Milk Protein Allergy"

_ijms, 2023, doi:10.3390/ijms241411422_

Round 1

Reviewer 1 Report

Thank you for your paper. Nice to read a well constructed paper. My concerns are relatively small

Overall HMO are known (and shown in multiple studies) to modify the fecal microbiome). The difference here is that the background is an extensively hydrolysed whey based formula.

The use of "shape" in the title feels like a marketing friven rather than scientific - suggest "modify" but this is optional. The title should reflect that this is only associated with early intervention not post complementary food introduction.

The dysbiosis associated with the extensive hydrolysed formula needs a bit of development in the introduction. 

Personally, found the EE and LE approach very logical so was looking for that to be carried through in all analyses but results got hard to follow due to the mixed approach. For example Figure 2 B and C do not align with the title of the figure as the FCT1-5 are a full experimental population analysis regardless of 2'FL/LNnT supplementation which confused me as a reader given the title says HMO modifying microbiome and metabolome, should not the FCT analysis be at least split to +/- 2'FL/LNnT if not to EE +/- 2'FL/LNnT and LE +/- 2'FL/LNnT

Fig 2A - As a reader I get suspicious when I look at data and 1/8 points in the diversity are significant. Probably this is correct as the other EE points are lower and 2F supports, but I don't find a clear point where you develop this properly so more work around line 450-8 would help the document. 12 month EE is not significant line 451 so don't lean too much on the "trend" as LE V1 has a similar "trend"

In the discussion I was looking for more development of the dysbiosis and metabolome  (aligning with the comment on the introduction) associated with extensively hydrolysed formula as the mechanism of 2'FL/LNnT may be different compared to infants without cows milk allergy

I have no concerns that would not be outside of normal editorial domain

Author Response

We would like to thank the reviewer for his/her thorough review and valuable comments. We have addressed all points raised and have detailed our response and actions taken below. We acknowledge that minor editing of English language was required and have had the manuscript proofread by native English speakers.

  • The use of "shape" in the title feels like a marketing driven rather than scientific - suggest "modify" but this is optional. The title should reflect that this is only associated with early intervention not post complementary food introduction.

Response:  We acknowledge that “shape” may not be the most appropriate wording for the title. We have changed the word by “modify”, as suggested by the reviewer.  We also thank the reviewer for the additional comment on the title, however, our results showed that the HMO combination impacted the gut microbiome in both cohorts, as described by a change in alpha diversity and a delay in the ecosystem maturation. The metabolomic analysis also showed distinct metabolic profiles for the control and HMO-treated groups in the EE as well as the LE cohort. These effects were indeed stronger for infants enrolled before 3 months, with greater enrichment of bifidobacteria and a clear impact on microbial amino acid and bile acid metabolism. As mentioned in the discussion, the HMO effect in the LE cohort might be more difficult to capture due to age variability and complementary diet. Thus, we think that the title should reflect the fact that, regardless of enrollment age, we observed an effect of HMOs on the microbiome and the metabolome of the infants with CMPA.

  • The dysbiosis associated with the extensive hydrolyzed formula needs a bit of development in the introduction

Response: We thank the reviewer for this point. Indeed, extensively hydrolyzed formulae contain different protein content than conventional formulae. We added a paragraph regarding the microbial changes linked with different protein content in the paper. We thought it was appropriate to include it in the discussion section where we also highlighted the dysbiosis linked with CMPA (lines 485-487)  

“In addition, since different protein content can affect the gut microbial ecosystem[38], the microbiome-mediated effect of HMO on infant health may be different in the context of EHF and reference formulae”

  • Personally, found the EE and LE approach very logical so was looking for that to be carried through in all analyses but results got hard to follow due to the mixed approach. For example Figure 2 B and C do not align with the title of the figure as the FCT1-5 are a full experimental population analysis regardless of 2'FL/LNnT supplementation which confused me as a reader given the title says HMO modifying microbiome and metabolome, should not the FCT analysis be at least split to +/- 2'FL/LNnT if not to EE +/- 2'FL/LNnT and LE +/- 2'FL/LNnT

Response: We thank the reviewer for highlighting this important point. The construction of FCT clusters was data-driven, and in order to compare the trajectories of corresponding FCTs between the test and control groups it was essential to define these clusters based on the entire population. Figures 2B and 2C show the taxonomical overview of each FCT cluster and the age-dependent appearance of FCTs (defining early and late FCTs, respectively). Since these characteristics were independent from the intervention or cohort, we did not stratify these results by EE and LE cohort. As suggested by the reviewer, we then compared the distribution of FCTs in the EE and LE cohorts, both with and without 2'-FL/LNnT, as shown in Figure 2D.

  • Fig 2A - As a reader I get suspicious when I look at data and 1/8 points in the diversity are significant. Probably this is correct as the other EE points are lower and 2F supports, but I don't find a clear point where you develop this properly so more work around line 450-8 would help the document. 12-month EE is not significant line 451 so don't lean too much on the "trend" as LE V1 has a similar "trend"

Response: We acknowledge the comment on statistical trend vs significance in the alpha diversity for the EE cohort. We have therefore removed the following statement in line 438 “in the EE cohort […] the same trend persisted to 12 month of age”.

  • In the discussion I was looking for more development of the dysbiosis and metabolome (aligning with the comment on the introduction) associated with extensively hydrolysed formula as the mechanism of 2'FL/LNnT may be different compared to infants without cow’s milk allergy

Response: This is valuable point from the reviewer. we have added the following section in the discussion:

“The pathophysiology of CMPA is complex and multifactorial [33]. Several studies have highlighted a dysbiotic gut microbiome in infants with CMPA characterized by a reduced microbial diversity, a loss of beneficial bacteria such as bifidobacteria, and the presence of opportunistic pathogens [34]” (lines 400-402)

“In our study, the microbial ecosystem of the EHF-fed infants with CMPA likely differed from that of healthy infants.[…] Nevertheless, we showed that HMO supplementation had a positive impact on the gut microbiome and its metabolism.” (Lines 484-490)

Reviewer 2 Report

This paper aimed to study the extensively hydrolyzed formula supplemented with 2’-FL and LNnT affects the infant’s fecal microbiome and metabolome with cow’s milk protein allergy, which was of general interest to International Journal of Molecular Sciences. Overall, the study was carried out rigorously, provides well-executed experiments with appropriate controls and the results are adequately presented and carefully discussed. However, the paper should be improved, previously to publication. I have listed some comments below:

1.       There are several types of oligosaccharides in breast milk, why did this study focus on exploring 2’-FL and LNnT?

2.       Why not level the amount of protein in the test formula and the control formula to the same level?

3.       Is the dosage of 2 '- FL and LNnT added calculated or obtained from other experiments?

4.       Line 652. Please declare all R packages used.

5.       Some references are missing page numbers, please correct.

6.       Table 1. Please change the table format to three-line table format.

7.       Figures 4 and 6 have a low resolution and cannot see the internal text details clearly. It is recommended to increase the resolution of the image.

Author Response

We would like to thank the reviewer for his/her valuable comments that helped to improve the content and the visuals of the paper.  For each comment, we have detailed our responses and actions below:

  • There are several types of oligosaccharides in breast milk, why did this study focus on exploring 2’-FL and LNnT?

Response: There are several reasons why the EHF was supplemented with 2'FL and LNnT. Of more than 130-150 HMO species, 2’-FL and LNnT were among the first breast milk-identical HMO manufactured by bacterial biofermentation that were approved for use in infant formulas. Both HMO are among the more abundant HMOs present in human milk. Data from a clinical trial in healthy infants fed an unhydrolyzed cow’s milk-based formula demonstrated health-related effects in infants, including a reduction in lower respiratory tract infections and antibiotic use in the first year of life (Puccio et al, 2017). This was reflected in specific changes in the faecal microbiome with an enrichment in bifidobacteria and associated metabolites. Based on this trial, 2’-FL and LNnT were deemed suitable bioactives for supplementation of hypoallergenic formulas in infants with cow’s milk protein allergy. This has been described as followed in the introduction (lines 68-73):

“A previous clinical trial has shown that non-hydrolysed cow’s milk-based infant formula supplemented with 2'-fucosyllactose (2’-FL) and lacto-N-neotetraose (LNnT) is well tolerated and supports normal growth in healthy term infants [13]. In that study, reduced incidences of lower respiratory tract infections (LRTI) and medication use (antipyretics and antibiotics) were reported in the study group fed with the HMO-supplemented infant formula, compared to the control group receiving no HMO [13]. The reduced antibiotic usage in the HMO-supplemented feeding group was associated with a shift of the gut microbiome composition closer to patterns observed in breastfed infants [14]. […] HMO supplementation in EHF formulae for CMPA may provide similar beneficial effect.”

  • Why not level the amount of protein in the test formula and the control formula to the same level?

Response:  The protein content in the test group reflected the amount present in the latest product. The reduced protein content is a trend in the development of modern infant formulas and aims to reduce the risk of excessive weight gain, compared to breastfed infants. This is described in the introduction of the first publication reporting the clinical outcomes of our clinical trial by Vandenplas et al, 2022 Nutrients.

"Apart from the overall dietary energy provided to infants, the daily protein intake was identified as an important modifier of weight gain and obesity risk in infants and children [23–25]. Higher levels of dietary protein in the first six months of life are associated with increased levels of branched-chain amino acids and acylcarnitines, which stimulate beta-oxidation and fat storage [26]. Over the past two decades, many standard infant formulas were reduced in protein content in an attempt to reduce the risk of excess weight gain [24]. By contrast, most extensively hydrolyzed formulas (EHF) and amino acid-based formulas (AAF) provide a relatively high protein equivalent in the form of peptides or free amino acids to counteract potential malabsorption and provide adequate nutrients for catch-up growth [27]. It is unclear whether the higher protein content in EHF and AAF could be associated with increased weight gain in infants with CMPA."

In addition, the following sentence has been added (line 602-604) in this publication:

“As described by Vandenplas et al, the HMO-containing infant formulae had slightly lower protein content reflecting the latest development in infant formulas, in attend to reduce risk of excess weight gain”.

  • Is the dosage of 2'- FL and LNnT added calculated or obtained from other experiments?

Response: Compliance to the target HMO concentrations, 2'FL 1g/L reconstituted formula and LNnT 0.5 g/L reconstituted formula, was verified by quantification of both HMOs. We now included in the manuscript that the values provided are the target values (lines 603, 604). The dose of HMOs was chosen based on previous clinical trial (Puccio, G et al, 2017: ref-13) in which positive health outcomes in healthy infants were found with 0.5g/L of LNnT and 1 g/L of 2’-FL.

  • Line 652. Please declare all R packages used

We have used the following R packages: vegan, picante, Dirichlet Multinomial, exactRankTests, effsize, ggplot2, ggrepel, igraph, Kendall, pheatmap, ComplexHeatmap, survival, survminer. The main text has been updated so all packages are now referenced within the respective method paragraphs. We chose this option instead of mentioning them all in Line 652.

In addition, we used several basic R packages, which are not referenced in the text, incl. ape, circlize, dplyr, forcats, ggarrange, glue, magrittr, openxlsx, patchwork, purrr, RColorBrewer, readxl, scales, stringr, tibble, and tidyr.

  • Some references are missing page numbers, please correct.

Response: We have added the page numbers where it was missing in the references

  • Table 1. Please change the table format to three-line table format.

Response: The table 1 has been changed accordingly

  • Figures 4 and 6 have a low resolution and cannot see the internal text details clearly. It is recommended to increase the resolution of the image

Response: We have increased the resolution of these figures to 300 dpi.